# Heterogeneous survival upon disinfection underlies evolution of increased tolerance

Niclas Nordholt,[1] Lydia-Yasmin Sobisch,[1] Annett Gödt,[1] Dominique Lewerenz,[1] Frank Schreiber[1]

**ABSTRACT** Disinfection is important to limit the spread of infections, but failure of disinfection may foster the evolution of antimicrobial resistance in bacteria. Persisters are phenotypically tolerant subpopulations that survive toxic stress longer than susceptible cells, leading to failure in treatments with antimicrobials and facilitating resistance evolution. To date, little is known about persistence in the context of disinfectants. The aim of this study was to investigate the influence of persisters on disinfection and to determine the consequences of disinfectant persistence for the evolution of increased tolerance to disinfectants. Disinfection kinetics with high temporal resolution were recorded for *Escherichia coli* exposed to the following six disinfectants: hydrogen peroxide ($H_2O_2$), glutaraldehyde (GTA), chlorhexidine (CHX), benzalkonium chloride (BAC), didecyldimethylammonium chloride (DDAC), and isopropanol (ISO). A mathematical model was used to infer the presence of persisters from the time–kill data. Time–kill kinetics for BAC, DDAC, and ISO were indicative of persisters, whereas no or weak evidence was found for $H_2O_2$, GTA, and CHX. When subjected to comparative experimental evolution under recurring disinfection, *E. coli* evolved increased tolerance to substances for which persisters were predicted (BAC and ISO), whereas adaptation failed for substances in which no persisters were predicted (GTA and CHX), causing extinction of exposed populations. Our findings have implications for the risk of disinfection failure, highlighting a potential link between persistence to disinfectants and the ability to evolve disinfectant survival mechanisms.

**IMPORTANCE** Disinfection is key to control the spread of infections. But the application of disinfectants bears the risk to promote the evolution of reduced susceptibility to antimicrobials if bacteria survive the treatment. The ability of individual bacteria to survive disinfection can display considerable heterogeneity within isogenic populations and may be facilitated by tolerant persister subpopulations. Using time–kill kinetics and interpreting the data within a mathematical framework, we quantify heterogeneity and persistence in *Escherichia coli* when exposed to six different disinfectants. We find that the level of persistence, and with this the risk for disinfection failure, depends on the disinfectant. Importantly, evolution experiments under recurrent disinfection provide evidence that links the presence of persisters to the ability to evolve reduced susceptibility to disinfectants. This study emphasizes the impact of heterogeneity within bacterial populations on disinfection outcomes and the potential consequences for the evolution of antimicrobial resistances.

**KEYWORDS** disinfection, persisters, tolerance, evolution, survival

**Peer Reviewer** Donald W. Schaffner, Rutgers, The State University of New Jersey, New Brunswick, New Jersey, USA

Address correspondence to Niclas Nordholt, niclas.nordholt@bam.de, or Frank Schreiber, frank.schreiber@bam.de.

Niclas Nordholt and Lydia-Yasmin Sobisch contributed equally to this article. Author order was determined both alphabetically and by seniority.

The authors declare no conflict of interest.

Disinfection is important to control microorganisms. However, incomplete killing with substances used for disinfection has been shown to facilitate the selection and evolution of bacteria with improved survival to the disinfectant to which they were exposed and, in addition, affecting antibiotic susceptibility of the bacteria evolved to the

disinfectant (1, 2). To minimize these unwanted side effects of disinfectants, it is crucial to understand their interaction with bacteria and the factors that interfere with their antimicrobial efficacy (2–5). An important factor that affects the efficacy of antimicrobial compounds is phenotypic heterogeneity (6, 7). Persister cells are a prime example of phenotypic heterogeneity affecting the outcome of antimicrobial treatment, with the potential to facilitate resistance evolution (8–10). Persister cells, which have been mainly studied in the context of antibiotics, are subpopulations which survive lethal antimicrobial stress much longer than the bulk of the population (11). The hallmark of persistence are bimodal time–kill kinetics when a bacterial population is challenged with a lethal dose of an antimicrobial (12). Recently, it has been shown that *Escherichia coli* forms persisters to the widely used disinfectant benzalkonium chloride (BAC) and that repeated persister-mediated failure of disinfection resulted in the evolution of BAC tolerance linked to changes in antibiotic susceptibility (1). For most disinfectants, knowledge on the extent of phenotypic heterogeneity affecting their antimicrobial activity and the resulting potential consequences for tolerance evolution is currently lacking.

Here, we sought to fill this knowledge gap by combining quantitative disinfection kinetics of *E. coli* with mathematical modeling and experimental evolution for a set of six chemically different and commonly used active substances used in disinfectants.

## MATERIALS AND METHODS

### Strains and growth conditions

Experiments were performed with *E. coli* K12 MG1655, which is widely used as model strain in persistence research (1, 8, 10, 13, 14). Bacteria were cultured in LB Lennox or M9 minimal medium with 20 mM glucose (M9) at 37°C with agitation at 220 rpm, as described earlier (1). LB Lennox (L3022, Sigma Aldrich) or M9 minimal medium (42 mM $Na_2HPO_4$, 22 mM $KH_2PO_4$, 8.5 mM NaCl, 11.3 mM $(NH_4)_2SO_4$, 1 mM $MgSO_4$, 0.1 mM $CaCl_2$, 0.2 mM uracil, 1 µg/mL of thiamine, trace elements [25 µM $FeCl_3$, 4.95 µM $ZnCl_2$, 2.1 µM $CoCl_2$, 2 µM $Na_2MoO_4$, 1.7 µM $CaCl_2$, 2.5 µM $CuCl_2$, 2 µM $H_3BO_3$] and 20 mM glucose) were used for bacterial cultures. Pre-cultures were inoculated from single-use −80°C freezer stocks into 10 mL of medium to a density of 10 colony-forming units (cfu)/mL for LB and $10^4$ cfu/mL for M9 and incubated at 37°C with agitation at 220 rpm for 24 h to stationary phase.

### Antimicrobial susceptibility testing

Minimum inhibitory concentrations (MIC) and minimum bactericidal concentrations (MBC) were determined in LB medium with a modified broth microdilution assay (15). The assay was modified by adjusting the disinfectant concentrations and adjusting the initial cell concentration to $10^9$ cfu/mL. Pre-cultures were adjusted to $10^9$ cfu/mL in 200 µL of LB Lennox containing increasing concentrations of disinfectant in 96-well polypropylene microplates (Greiner) and incubated for 24 h at 37°C with shaking. Growth was assessed in an Epoch microplate reader (Biotek). Because the time–kill assays were conducted at a cell density of $10^9$ cfu/mL, the MIC and MBC assays were adjusted accordingly to account for the inoculum effect (4, 16, 17), in which the interaction of the biomass with the substances can minimize their efficacy. The lowest concentration of disinfectant inhibiting growth was designated the MIC (15). The lowest concentration, which reduced the initial cell number by 99.999%, was designated the MBC, as most guidelines require disinfectants to exhibit a reduction in the initial cell number by 99.999% to be deemed bactericidal [(18), pp. 401–419] (19). In line with this, the MBC here refers to a viable cell reduction of 99.999%. Concentrations tested in the MIC and MBC assays were as follows: $H_2O_2$ [mM]: 0.39, 0.78, 1.56, 3.125, 6.25, 12.5, 25, 50, 100, 200; glutaraldehyde [%]: 0.02, 0.04, 0.08, 0.16, 0.23, 0.31, 0.47, 0.63, 0.94, 1.25; chlorhexidine [µg/mL]: 2, 3, 4, 5, 6.25, 9.375, 12.5, 18.75, 25, 35; benzalkonium chloride [µg/mL]: 5.3, 7.1, 14.2, 17.7, 21.2, 24.8, 28.3, 31.8, 35.4, 53.1; DDAC [µg/mL]: 14, 16, 24, 32, 40, 48, 56, 64, 72, 80; isopropanol [%]: 0.07, 0.14, 0.275, 0.55, 1.1, 2.2, 4.4, 8.75, 17.5, 35.

## Determination of time–kill kinetics

Time–kill kinetics at high temporal resolution were performed with *E. coli* pre-cultured in LB medium to reach stationary phase, as described earlier (1), with the modifications wherein the initial cell density was adjusted to $10^9$ cfu/mL, and killing assays were performed in phosphate-buffered saline (PBS). Time–kill assays were carried out with *E. coli* populations in stationary phase, when persister formation is induced (8, 13, 14). Pre-cultures were harvested by centrifugation at 4,000 *g* for 4 min and adjusted to $4 \times 10^9$ cfu/mL in PBS (2.68 mM KCl, 1.76 mM $KH_2PO_4$, 10 mM $Na_2PO_4$, 137 mM NaCl). Cell suspensions were then diluted to a final concentration of $10^9$ cfu/mL in PBS, and disinfectant was added to initiate killing in a final volume of 900 µL, followed by incubation in a ThermoMixer (StarLab) at 37°C with agitation at 1,200 rpm. Prior to and during killing, cfu were determined by sampling 10 µL, serial dilution in PBS, and spotting onto LB agar plates. Disinfectant concentrations were chosen to be in the range of the MBC and are listed in Table 1. To test whether resistant mutants were responsible for wide tolerance distributions with prolonged survival for >20 min observed for DDAC and CHX, two colonies originating from cells, which survived for 15 min or longer were picked and subjected to MIC determinations as described above. A colony, which has not been exposed to disinfectant, served as a control. No differences in the MICs between control and late colonies were observed. To preclude that exhaustion of disinfectant from the medium caused the observed kinetics, $10^8$ cfu/mL were spiked into a killing assay with CHX after 22 min, and similar kinetics were observed (Fig. S1). Both assays were previously performed for BAC (1).

## Experimental evolution and calculation of evolvability score

Bacteria were cultured in M9 for 24 h and subjected to killing for a duration that reduced the cfu by a factor of $10^4$ to $10^5$ based on data from Fig. 1. After treatment for 2 (GTA), 5 (ISO), or 10 min (CHX), 100 µL were sampled and diluted in 10 mL of fresh M9 to restart the cycle. The number of surviving cells was monitored by serial dilution and spotting on LB agar plates. To compare the evolution experiments with different substances, we devised an evolvability score that considers the number of replicate populations, which survive the evolution experiment, the time needed to adapt, and the extent of adaptation in terms of increased survival. Evolvability scores were calculated for each replicate population. For this, the survival fraction at each transfer was normalized by the initial survival fraction. These data were then log transformed, and the area under the curve (AUC) was approximated via the trapezoidal rule (20) according to the following formula:

**TABLE 1** Overview of the disinfectants used and their antimicrobial properties

| Disinfectant [units] | Minimum inhibitory concentration (MIC)[a] | Minimum bactericidal concentration (MBC)[a] | Concentration in time–kill and evolution assay | Predicted persister fraction g[b] |
|---|---|---|---|---|
| Hydrogen peroxide ($H_2O_2$) [mM] | 12.5–25 | 12.5–25 | 75 | 0 |
| Glutaraldehyde (GTA) [%] | 0.08–0.16 | 0.08–0.16 | 0.015[c] | 0 |
| Chlorhexidine (CHX) [µg/mL] | 6.25–9.375 | 18.75–25 | 100 | 0 |
| Benzalkonium chloride (BAC) [µg/mL] | 31.8–35.4 | 35.4–53.1 | 42.5 | $3 \times 10^{-3}$ $[2 \times 10^{-3} – 6 \times 10^{-3}]$ |
| Didecyldimethyl-ammonium chloride (DDAC) [µg/mL] | 16–24 | 16–24 | 28 | $1.4 \times 10^{-4}$ $[3.6 \times 10^{-5} – 5 \times 10^{-4}]$ |
| Isopropanol (ISO) [%] | 2.2–4.4 | 4.4–8.75 | 12 | $7.6 \times 10^{-4}$ $[2.5 \times 10^{-4} – 2.3 \times 10^{-3}]$ |

[a]Concentrations tested in the MIC and MBC assays: $H_2O_2$ [mM]: 0.39, 0.78, 1.56, 3.125, 6.25, 12.5, 25, 50, 100, 200; glutaraldehyde [%]: 0.02, 0.04, 0.08, 0.16, 0.23, 0.31, 0.47, 0.63, 0.94, 1.25; chlorhexidine [µg/mL]: 2, 3, 4, 5, 6.25, 9.375, 12.5, 18.75, 25, 35; benzalkonium chloride [µg/mL]: 5.3, 7.1, 14.2, 17.7, 21.2, 24.8, 28.3, 35.4, 53.1; DDAC [µg/mL]: 14, 16, 24, 32, 40, 48, 56, 64, 72, 80; isopropanol [%]: 0.07, 0.14, 0.275, 0.55, 1.1, 2.2, 4.4, 8.75, 17.5, 35.
[b]Parameter values for best model fits are given in Table S1 . The range indicates the upper and lower limit of the calculated persister fraction.
[c]The lower concentration in the time–kill assay compared to the MBC for GTA is explained by the strong effect of organic matter in LB used for MBC determination compared to phosphate-buffered saline used for the time–kill assay.

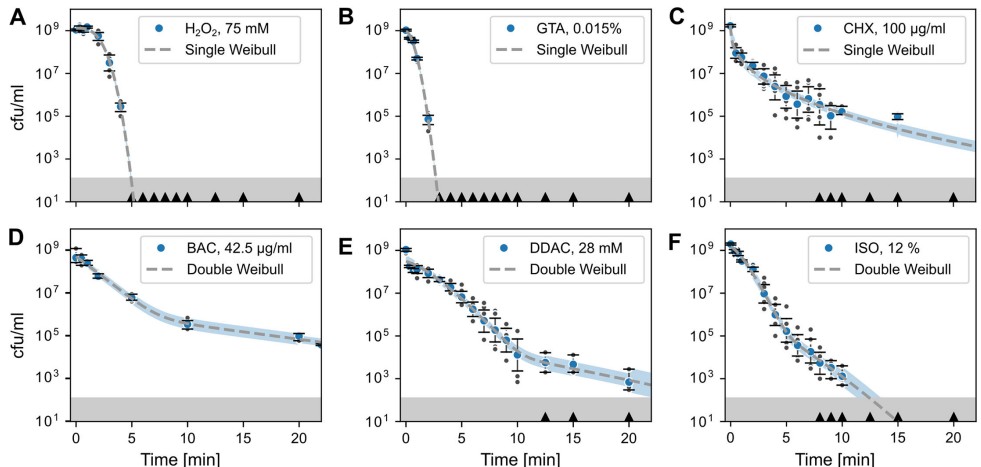

**FIG 1** Phenotypic heterogeneity affects disinfection kinetics. Disinfection time–kill kinetics of *E. coli* exposed to different disinfectants and fit by Weibull distributions (dashed line). Data in panels (A) to (C) are best fitted with a single Weibull model (Equation 2), whereas data in panels (D) to (F) are best fitted with a double Weibull model (Equation 3). Blue circles indicate the geometric mean of the time–kill experiments; error bars indicate the 95% CI obtained by bootstrapping. Dark gray circles are datapoints of individual experiments (in some panels, individual data points are overlayed by the data point displaying the geometric mean). Black triangles on the x-axes indicate when zero counts were present. Blue shaded areas indicate the 95% CI of the model fit (dashed line) to the experimental data, excluding values with zero counts. The gray shaded area at the bottom indicates the detection limit. Number of biological replicates $n = 6$, except for $H_2O_2$ where $n = 5$ and BAC where $n = 3$ biological replicates. (A) $H_2O_2$; (B) GTA; (C) CHX; (D) BAC; (E) DDAC; (F) ISO. Source data are provided in Dataset S1.

$$\text{AUC} = \frac{1}{2}\left(f(t_1) + 2f(t_2) + 2\,f(t_3) + \ldots + f(t_n)\right) \qquad (1)$$

where $t_n$ is the *n*-th transfer and $f(t_n)$ is the log-transformed, normalized survival fraction at transfer $t_n$. An *ad hoc* penalty of 1 was subtracted from the AUC for each transfer that was missed due to population extinction up to the 11th cycle to account for differences in the duration of disinfection survival of populations between different compounds.

## Calculations and statistics

A phenomenological mathematical model based on the sum of two Weibull distributions was used to infer whether time–kill kinetics were best explained by the presence of a tolerant subpopulation (21). The model assumes that a distribution of tolerance times underlies the disinfection kinetics, and it can account for the non-log-linear time–kill kinetics observed during disinfection (Fig. 1; Fig. S2). The model describes the number of survivors *N* consisting of two subpopulations as function of time *t* as:

$$N(t) = \frac{N_0}{1 + 10^a}\left(10^{-\left(\frac{t}{d_1}\right)^p + a} + 10^{-\left(\frac{t}{d_2}\right)^p}\right) \qquad (2)$$

where $N_0$ is the inoculum size in cfu/mL, *a* is the logit transformed fraction *f* of the susceptible population ($a = \log\frac{f}{1-f}$), *p* is a shape parameter, and $d_1$ and $d_2$ are the treatment times for the first decimal reduction of population 1 (susceptible) and 2 (persister), respectively. The persister fraction *g* is yielded by $g = \frac{1}{10^a + 1}$. For simplicity, the shape parameter *p* is set to be the same for both Weibull distributions (21). When only one population is present, the number of survivors can be described by a single Weibull distribution:

$$N(t) = N_0 10^{-\left(\frac{t}{d_1}\right)^p} \qquad (3)$$

Further details on the model fitting algorithm and the derivation of the model are given in Text S1 and S2, respectively. Equations 2 and 3 were fit to the log-transformed data of colony counts of all biological replicates, using the lmfit package for Python 3.8 (22). The Akaike information criterion (AIC) was used to choose the best model and the robustness of the fitting method to the measurement noise, and parameter dependence was assessed in depth (Fig. S3, Text S3).

## RESULTS

### Time–kill kinetics reveal heterogeneous tolerance to disinfectants

First, we investigated the phenotypic heterogeneity in survival upon disinfection by conducting time–kill assays. We used six different substances belonging to the classes of quaternary ammonium compounds (BAC and DDAC), alcohols (ISO), aldehydes (GTA), oxidative substances ($H_2O_2$), and cationic biguanides (CHX). As a reference for the concentrations used in the time–kill assays, MIC and MBC were determined (Table 1). The shape of the time–kill kinetics was dependent on the disinfectant (Fig. 1). Fitting a single (unimodal) and a double (bimodal) Weibull model to the data showed no evidence for a persister subpopulation for $H_2O_2$ and GTA (Fig. 1A and B). The wide disinfection kinetics for CHX with prolonged survival for >20 min indicated high heterogeneity of the tolerance times of individual cells (Fig. 1C). However, there was no need to invoke a persister subpopulation to explain the long-tailed time–kill curve (Table 1), as the unimodal model provided a slightly better fit (Table S1).

The time–kill kinetics for BAC, DDAC, and ISO were best explained by a bimodal Weibull model, suggesting the presence of a tolerant persister subpopulation (Fig. 1D through F; Table 1). The results for BAC corroborate findings from a previous in-depth study conducted by us, under slightly different pre-culture and killing conditions (1). Resistant mutants were excluded to cause the wide tolerance time distributions with prolonged survival for >20 min against DDAC and CHX by showing that the MIC of colonies from the 15-min time-point remained unchanged (data not shown) compared to the MIC displayed in Table 1. Furthermore, it was tested whether a loss of disinfectant activity over the course of the assay was responsible for the long-tailed kinetics of CHX. To this end, fresh cells were spiked into the late phase of a time–kill assay, which resulted in similar disinfection kinetics (Figure S1). Both assays were previously conducted for BAC with similar results (1). Together, these data suggest that tolerant persister subpopulations affect disinfection kinetics in a substance-dependent manner.

### Persister subpopulations facilitate tolerance evolution

Repeated, persister-mediated failure of disinfection has been shown to rapidly select for BAC tolerance (1). To investigate the role of persistence in the evolution of disinfectant tolerance, *E. coli* populations were evolved under recurrent exposure to selected disinfectants with unimodal (GTA, CHX) or bimodal (ISO) time–kill kinetics at a specific concentration that reduces viability to a comparable degree. The results of the evolution experiments (Fig. 2A, B and D) were compared to the results of a similar, previously published evolution experiment with BAC (Fig. 2C) (1). Bacterial populations were exposed to disinfectant concentrations that reduced viability by $10^2$ to $10^4$ within 2 to 15 min, followed by dilution and growth in fresh medium for up to 11 cycles. Remarkably, all populations treated with disinfectants, which exhibited unimodal time–kill kinetics (GTA, CHX) at the chosen concentration, failed to adapt to the treatment and went extinct during the experiment, even when occasional temporary increases in survival were observed (Fig. 2A, B and E). In contrast, populations that were treated with disinfectants exhibiting bimodal time–kill kinetics (ISO, BAC) at the chosen concentration adapted to the treatment through increased levels of survival (Fig. 2C, D and E). To assess the degree of adaption in relation to the persister fraction obtained by the model (Fig. 1;

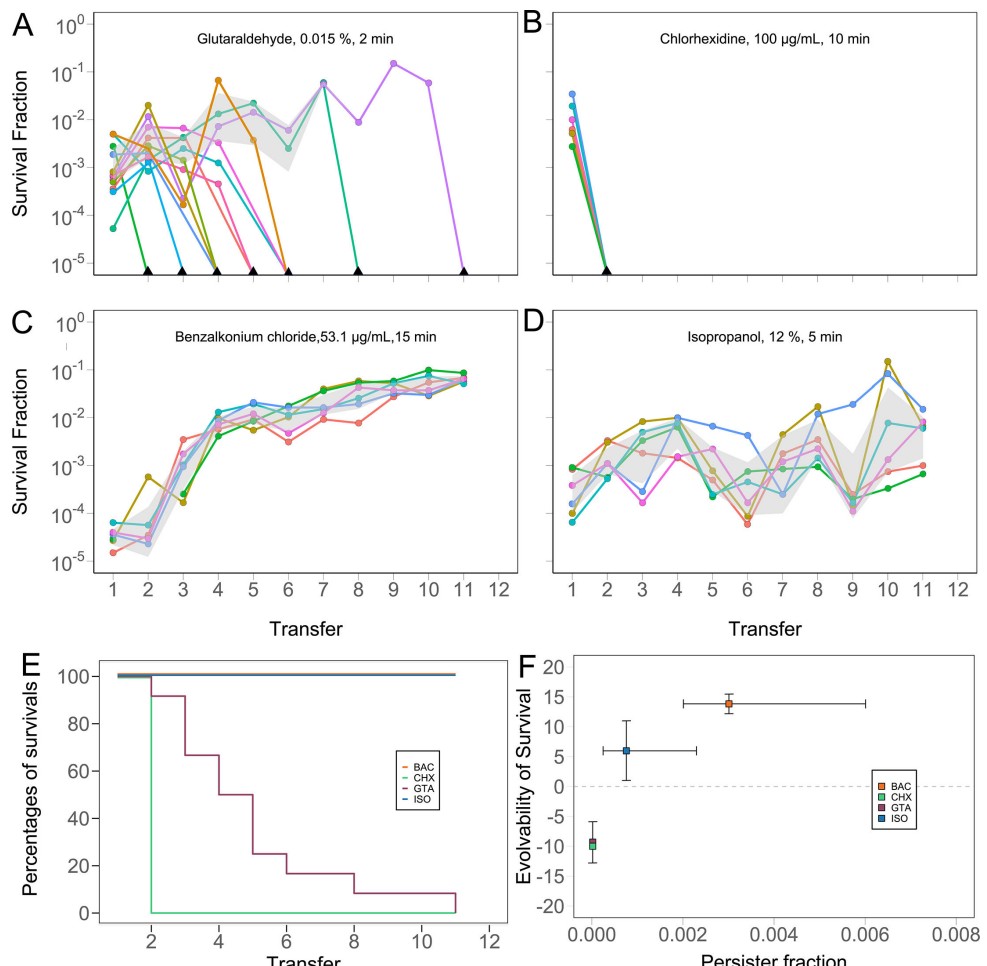

**FIG 2** Tolerance evolution in *E. coli* under periodic treatment with disinfectants is related to the level of persistence. Evolution of survival in populations repeatedly exposed to (A) glutaraldehyde (GTA), (B) chlorhexidine (CHX), (C) benzalkonium chloride (BAC; data taken from [1]), and (D) isopropanol (ISO). Number of replicate evolutionary lineages ($n = 6$), except for GTA ($n = 12$). The gray area shows the 95% CI, excluding zero values. (E) Fraction of surviving replicates of BAC, ISO, GTA, and CHX over 11 treatment cycles. Absence of survival of replicates implies the inability to evolve tolerance in terms of the ability to re-grow in liquid medium after disinfectant exposure. (F) Relationship between the ability to evolve tolerance against the disinfectant and the persister fraction in the ancestor population. The evolvability score was calculated as described in the "Materials and Methods" section. The persister fraction $g$ was calculated with $g = \frac{1}{10^a + 1}$, where $a$ was obtained from the fit of Equation 2 to data in Fig. 1 (cf. Table S1). Error bars indicate the standard deviation across replicates (evolvability of survival) and the 95% CI on the predicted persister fraction. Source data are provided in Dataset S2.

Table 1), evolvability scores were calculated for each population (Fig. 2F). The evolvability score considers the number of replicate populations, which survive the treatment, the time needed to adapt, and the extent of adaptation in terms of increased survival. A positive evolvability score indicates adaptation toward the tested disinfectant through increased survival, a score of zero indicates no change in survival, i.e., no adaptation, and a negative score indicates a decrease in survival or extinction events, i.e., an outcome worse than expected when no adaptation occurs. The data suggest a positive association between the size of the persister subpopulation and the evolvability score (Fig. 2F), but additional data for other disinfectants are required to further substantiate this relationship for the *E. coli* strain used in this study. Taken together, *E. coli* adapted to disinfectants at concentrations leading to bimodal time–kill kinetics but failed to adapt

to disinfectants at concentrations leading to unimodal kinetics, suggesting that in these experiments, persisters facilitated the evolution of disinfectant tolerance.

## DISCUSSION

Antibiotic persistence has been associated with treatment failures and the evolution of antibiotic resistance (10–12), but little is known about the impact of persistence on disinfection and disinfectant tolerance evolution. Here, we identified several disinfectants that exhibited time–kill kinetics indicative for the presence of persister cells in *E. coli*. Through comparative experimental evolution, we observed that tolerance evolved under conditions where persisters were present, whereas adaptation failed in their absence.

Our results suggest that phenotypic heterogeneity in the form of disinfectant tolerant persister subpopulations is a prevalent phenomenon. Similarly, antibiotic persistence appears to be the rule, rather than the exception, considering that persistence has been shown across a multitude of antibiotics (12, 23). Several known antibiotic persistence mechanisms in *E. coli* have been shown to overlap with survival mechanisms to the disinfectant benzalkonium chloride (1). However, a main difference between classical antibiotic persisters and the disinfectant persisters, which we observe here is their ability to survive application concentrations of antibiotics and disinfectants. The disinfectant concentrations that were used in our assays inactivated high-density cultures within short timeframes, yet these concentrations are at the lower end of the application concentrations of the tested substances (Table S1). However, our assays were conducted in the absence of organic matter, i.e., soiling, which hampers a direct comparison of application concentrations and the concentrations used here. Soiling can diminish the activity of disinfectants and is usually accounted for in application concentrations ((18), pp. 401–419). Importantly, we cannot exclude that there are concentrations or conditions that would also expose persistence to the substances that did show unimodal inactivation kinetics in our experiments (GTA, $H_2O_2$, CHX). Disinfection kinetics, and in general chemical inactivation kinetics, are concentration dependent (1, 24–27). However, the situation at or around concentrations that inactivate at least 99.999% of bacteria, as is required for disinfectants (18), is not well studied, especially not in the light of persistence and evolvability of increased survival. Therefore, future experiments should focus on these aspects to understand and prevent failure of disinfection and evolution of increased survival upon disinfection.

The mechanistic basis of prolonged survival upon disinfection is not clear, but as for antibiotics, it is likely multifactorial and dependent on the mode-of-action, the bacterial species, the concentration, and the specificity and multitude of the cellular targets. Notably, all cationic substances assayed here exhibited wide disinfection kinetics with prolonged survival for >20 min (Fig. 1), which may be attributable to their specific mode of action, compared to very unspecific agents like glutaraldehyde or hydrogen peroxide for which no survival was detected after 5 min. The target of cationic antiseptics is the cell envelope (28). Therefore, modifications to the envelope, e.g., through altered membrane charge, or species-specific differences (Gram-positive versus Gram-negative), can alter susceptibility to these compounds and therefore disinfection kinetics (1, 29–31). Furthermore, these substances can be subject to multi-drug efflux (32, 33). Recently, it has been found that mutations related to metabolism can induce pan-tolerance toward antibiotics and disinfectants (2). This suggests that phenotypic heterogeneity and maladaptation of individual cells in terms of surface charge, efflux pump activity, and metabolism may contribute to disinfection persistence. These mechanisms are known antibiotic persistence mechanisms and have been shown to modulate persistence to benzalkonium chloride (1). Single-cell studies are required to gain deeper mechanistic insights, for instance, by combining fluorescent disinfectant analogs with time-lapse microscopy (32). Single-cell survival times can then be related to population-level time–kill kinetics, using the Weibull model employed in this study (21, 34).

The interpretation of time–kill curves as probability distributions highlights the heterogeneity within isogenic populations, capturing biological phenomena such as persistence and making them immediately accessible to probabilistic modeling approaches (34, 35). The Weibull model has previously been used to investigate physical inactivation kinetics of microorganisms due to its ability to account for linear and non-linear inactivation kinetics, yet its application in modeling chemical inactivation is limited (21, 34, 36, 37). An extension of the model to account for strength of the lethal stress, e.g., different disinfectant concentrations, has been discussed (35). These properties make the Weibull model suitable to model time–kill kinetics of other bacterial species, a range of different chemical substances, including disinfectants and antibiotics, and across wider concentration ranges.

The evolution experiments conducted here corroborate earlier findings that persistence can facilitate adaptive evolution toward antibiotics and disinfectants (1, 10, 38). Persisters are thought to contribute to higher evolvability not only through creating the opportunity for mutations by survival of the population but also through elevated mutation rates, which may be the result of stress-induced mutagenesis (9, 39–41). The results of our evolution experiments support this view because adaption was only observed toward disinfectants for which persistence was predicted at the employed concentrations (Fig. 2E). In line with our observation that *E. coli* evolved tolerance toward isopropanol, a recent study found evidence that *Enterococcus faecium* isolates had become more tolerant toward isopropanol at a concentration of 23% after the introduction of alcohol-based hand rubs in a hospital (42).

We applied an evolution protocol of recurrent disinfection with short exposure to lethal concentrations of disinfectant, selecting for increased survival, i.e., tolerance. This protocol is intended to mimic disinfection and differs from protocols that select for growth in increasing concentrations of antimicrobials. Therefore, comparison of the results of our evolution experiments for chlorhexidine and glutaraldehyde with earlier works, which have shown adaptation to sub-lethal concentrations of chlorhexidine or glutaraldehyde (29, 30, 43), is complicated by the distinct evolution protocols and the different adaptation mechanisms responsible for survival upon exposure to lethal antimicrobial concentrations and growth in the presence of lower concentrations of antimicrobials (1, 2, 44). Future studies should focus on the identification of survival mechanisms to disinfectants and substantiating the here-observed relation between persistence and evolvability and extend them to further disinfectant active substances and products, wider concentration ranges for a particular substance, and additional relevant microbial species.

## Conclusions

In this study, we find evidence that phenotypic heterogeneity can affect disinfection kinetics, namely, through the formation of tolerant persister subpopulations, which exhibit prolonged survival upon exposure to lethal disinfectant concentrations. The extent of heterogeneity, and with this the risk for disinfection failure, depends on the disinfectant. Our results highlight substances for which mechanisms of phenotypic tolerance exist, which may have consequences for the evolution of population-wide tolerance when disinfectants are applied under conditions that do not eradicate the entire population. Such incomplete eradication is covered by international standards that define the appropriate efficacy of disinfectants to ensure a particular killing in terms of log reduction (typically 4 to 5 logs) (18), but do not require eradication below the lowest possible limit of detection (i.e., close to full eradication) (45). Thus, disinfectants and antiseptics for which bacterial populations contain persister cells might be at risk to evolve increased survival after multiple exposure cycles even if applied correctly. One barrier for such evolution in the field might be the prerequisite that populations need to regrow in the absence of growth-inhibitory stress to allow for the supply of a sufficient number of mutants for tolerance selection. Safeguarding the efficacy of disinfectants requires further research into more bacterial species, the physiology of

disinfectant persisters, their occurrence under more realistic disinfection scenarios, their role for evolvability of increased survival, the physiology of evolved disinfectant tolerant mutants also in terms of antibiotic susceptibility, how cycling of disinfectants impacts evolution, and potential barriers for evolution in the field.

## ACKNOWLEDGMENTS

This work was supported by the Federal Institute for Materials Research and Testing within the funding scheme "Menschen Ideen–Typ 1" (#MIT1-19-17).

## AUTHOR AFFILIATION

[1]Division of Biodeterioration and Reference Organisms (4.1), Department of Materials and the Environment, Federal Institute for Materials Research and Testing (BAM), Berlin, Germany

## AUTHOR ORCIDs

Niclas Nordholt http://orcid.org/0000-0001-5788-0801
Lydia-Yasmin Sobisch http://orcid.org/0009-0009-3409-5777
Frank Schreiber http://orcid.org/0000-0003-1957-6328

## AUTHOR CONTRIBUTIONS

Niclas Nordholt, Conceptualization, Data curation, Formal analysis, Investigation, Methodology, Software, Supervision, Validation, Visualization, Writing – original draft, Writing – review and editing | Lydia-Yasmin Sobisch, Data curation, Formal analysis, Investigation, Methodology, Validation, Visualization, Writing – original draft, Writing – review and editing, Conceptualization, Supervision | Annett Gödt, Formal analysis, Investigation, Methodology, Writing – review and editing, Data curation | Dominique Lewerenz, Data curation, Formal analysis, Investigation, Methodology, Writing – review and editing | Frank Schreiber, Conceptualization, Funding acquisition, Project administration, Resources, Supervision, Formal analysis, Methodology, Writing – original draft, Writing – review and editing

## DATA AVAILABILITY

The data generated in this study are enclosed with the manuscript (Datasets S1 and S2).

## ADDITIONAL FILES

The following material is available online.

### Supplemental Material

**Supplemental material (Spectrum03276-22-s0001.pdf).** Fig. S1 to S3; Table S1; Methods.
**Dataset S1 (Spectrum03276-22-s0002.xlsx).** Source data for Figure 1.
**Dataset S2 (Spectrum03276-22-s0003.xlsx).** Source data for Figure 2.

### Open Peer Review

**PEER REVIEW HISTORY (review-history.pdf).** An accounting of the reviewer comments and feedback.

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
