## [Reviewer comments · Microbiology Spectrum]

Microbiology Spectrum

Heterogeneous survival upon disinfection underlies evolution of increased tolerance

Niclas Nordholt, Lydia-Yasmin Sobisch, Annett Gödt, Dominique Lewerenz, and Frank Schreiber

Corresponding Author(s): Frank Schreiber, Federal Institute For Materials Research and Testing

Review Timeline:

Submission Date:	July 15, 2024
Editorial Decision:	August 21, 2024
Revision Received:	September 20, 2024
Accepted:	September 24, 2024

Editor: Olaya Rendueles

Reviewer(s): Disclosure of reviewer identity is with reference to reviewer comments included in decision letter(s). The following individuals involved in review of your submission have agreed to reveal their identity: Donald W. Schaffner (Reviewer #1)

Transaction Report:

DOI: <https://doi.org/10.1128/spectrum.03276-22>

Re: Spectrum03276-22 (Heterogeneous survival of disinfection underlies evolution of increased tolerance)

Dear Dr. Schreiber, Dear Frank

Your manuscript has been thoroughly modified following the reviewers comments, it is much clearer and the addition of new experiments has improved significantly this manuscript. One of the previous reviewers has also read the manuscript again and they are now positive and appreciate all the modifications included in this version.

Below you will find instructions from the Spectrum editorial office, and the reviewer comments.

Revision Guidelines

Sincerely,
Olaya Rendueles
Editor
Microbiology Spectrum

Reviewer #1 (Comments for the Author):

The authors present an interesting manuscript on evolution of E. coli resistance to various disinfectants. I have a number of minor comments relating to the inclusion of methods in the results section, as well as minor comments about the appropriateness of certain references.

My major comments are about the authors statements about bimodal inactivation kinetics. I wonder whether inactivation kinetics are dependent on the concentration of the anti-microbial agent, and I would encourage the authors to more fully explore this idea or provide evidence that it is incorrect.

Page 2:

Content: "leading to unexpected failure of treatments"

Comment: I'm not sure that it's unexpected given evolution.

Page 2:

Content: "known about persistence in the context of disinfectants."

Comment: I'm not exactly sure what the authors mean. The development of resistance to disinfectants is relatively well studied.

Page 4:

Content: "disinfectants and antibiotics [1], [2]."

Comment: Both of these citations are about benzalkonium chloride specifically not disinfectants generally. Neither of them are about antibiotics. I'm not sure that bringing antibiotics into the mix is especially helpful.

Page 4:

Content: "antimicrobial efficacy [3]."

Comment: Again, this is another specific reference about benzalkonium chloride

Page 5:

Content: "The lowest concentration which reduced the initial cell number by factor 105 was designated the MBC."

Comment: Is this the typical criterion for MBC? Can the authors provide a citation?

Page 7:

Content: "For this, the survival fraction at each transfer was normalized by the initial survival fraction. This data was then log-transformed and the area under the curve (AUC) was approximated via the trapezoidal rule."

Comment: Citation?

Page 7:

Content: "penalty of 1 was subtracted from the AUC for each transfer that was missed due to extinction, up to the 11th cycle."

Comment: Citation?

Page 8:

Content: "The goal of this study was to investigate the influence of phenotypic heterogeneity on disinfection and on the evolution of disinfectant tolerance."

Comment: Most of the information in the paragraph that begins with the sentence is not results. It's either method or it recapitulates the introduction. It should be deleted from the results.

Page 10:

Content: "Time-kill assays were carried out with E. coli populations in stationary phase, when persister formation is induced [6], [11], [12]."

Comment: Methods not results

Page 10:

Content: "To infer whether the observed time-kill kinetics were best explained by the presence of a tolerant persister subpopulation [10], a single (unimodal) and a double (bimodal) Weibull model were fitted to the data. The Weibull model has previously been used to model thermal inactivation of bacteria and can accommodate non-log-linear time-kill kinetics [14], [18], [19]. The model that best explained the observed data was selected based on goodness of fit and parsimony of the model (details in Text S1)."

Comment: This is all methods, not results

Page 10:

Content: "Overall, the killing with ISO proceeded faster than with the quaternary ammonium compounds BAC and DDAC."

Comment: This is true, but isn't all of this also impacted by the concentration of the antimicrobial? If lower concentrations were used, would the inactivation rate would be lower?

Page 10:

Content: "MIC of colonies from the 15 minutes time-point remained unchanged compared to the reference value (Table 1)."

Comment: I don't see where the MIC of colonies from the 15 minute time point or reference values are shown in table 1.

Page 11:

Content: "To investigate the role of persistence in the evolution of disinfectant tolerance, E. coli populations were evolved under

recurrent exposure to selected disinfectants with unimodal (GTA, CHX) or bimodal (ISO) time-kill kinetics."
Comment: Methods not results

Page 11:

Content: "Bacterial populations were exposed to disinfectant concentrations that reduced viability by 10² to 10⁴ within 2 to 15 minutes, followed by dilution and growth in fresh medium for up to 11 cycles."
Comment: Methods not results

Page 11:

Content: "To assess the degree of adaption in relation to the persister fraction obtained by the model (Figure 1; Table 1), evolvability scores were calculated for each population (Figure 2 F)."
Comment: As written, this sentence is methods not results

Page 12:

Content: "but additional data for other disinfectants are required to further substantiate this relationship."
Comment: It would also be good to examine this with other microorganisms.

Page 12:

Content: "E. coli adapted to disinfectants with bimodal time-kill kinetics but failed to adapt to disinfectants with unimodal kinetics, suggesting that in these experiments, persisters facilitated the evolution of disinfectant tolerance"
Comment: I get what the authors are saying here, but it seems to me that bimodal inactivation kinetics are evidence that a resistant subpopulation already exists. Is it possible for those situations where there's unimodal inactivation kinetics that this population does not exist because the concentration of the antimicrobial is too high. Is there any relationship between concentration of the antimicrobial and the inactivation kinetics?

Page 12:

Content: "results suggest that phenotypic heterogeneity in the form of disinfectant tolerant persister subpopulations is more prevalent than previously thought. "
Comment: Previously thought by who? Can the authors supply citation?

Page 12:

Content: "considering that persistence has been shown across a multitude of antibiotics [10], [20]."
Comment: In fact, I believe there is evidence of antibiotic resistance in populations of bacteria, recovered from samples that predate the discovery of antibiotics.

Page 13:

Content: "Soiling can diminish the activity of disinfectants and is usually accounted for in application concentrations."
Comment: A citation here would be good.

Page 13:

Content: "survival of disinfection"
Comment: Survival of disinfection seems like an odd phrase. It's not the disinfection that is surviving. It is the microorganisms.

Page 13:

Content: "dependent on the mode-of-action, the bacterial species, and the specificity and multitude of the cellular targets."
Comment: Probably also dependent on the concentration of the antimicrobial.

Page 13:

Content: "wide disinfection kinetics"
Comment: Wide disinfection kinetics seems an imprecise phrase. Can the authors be more specific about what they mean?

Page 14:

Content: "a recent study found evidence that E. faecium isolates had become more tolerant towards isopropanol after the introduction of alcohol-based hand rubs in health care facilities [34]."
Comment: While this is a correct summary of the authors findings, it is important to note that these authors used a concentration of isopropanol (23%), which is well below the recommended concentration.

Page 15:

Content: "particular killing in terms of log reduction (typically 4 to 5 logs), but do not require full eradication [37]."
Comment: There is no such thing as "full eradication". There is only inactivation to below detection limits. This depends on the starting concentration, the inactivation kinetics, and the detection limit.

Page 15:

Content: "Safeguarding the efficacy of disinfectants requires"

Comment: One commonly used method to prevent development of resistance is to change disinfectants to another disinfectant with a different mode of action. Therefore, it would also be useful to study how sequential exposure to different disinfectant impacts the development of resistance evolution.

Reviewer #1 (Comments for the Author):

The authors present an interesting manuscript on evolution of E. coli resistance to various disinfectants. I have a number of minor comments relating to the inclusion of methods in the results section, as well as minor comments about the appropriateness of certain references.

My major comments are about the authors statements about bimodal inactivation kinetics. I wonder whether inactivation kinetics are dependent on the concentration of the anti-microbial agent, and I would encourage the authors to more fully explore this idea or provide evidence that it is incorrect.

Page 2:

Content: "leading to unexpected failure of treatments"

Comment: I'm not sure that it's unexpected given evolution.

Response: The word 'unexpected' was removed.

Page 2:

Content: "known about persistence in the context of disinfectants."

Comment: I'm not exactly sure what the authors mean. The development of resistance to disinfectants is relatively well studied.

Response: This was maybe a misunderstanding. We stated that relatively little is known in the context of persistence to disinfectants, not resistance as stated by the reviewer. We agree that resistance to disinfectants has been studied relatively well, but persistence has not.

Page 4:

Content: "disinfectants and antibiotics [1], [2]."

Comment: Both of these citations are about benzalkonium chloride specifically not disinfectants generally. Neither of them are about antibiotics. I'm not sure that bringing antibiotics into the mix is especially helpful.

Response: We acknowledge that the sentence might be misleading and reformulate it accordingly. Our aim was to highlight that the effects on antibiotic susceptibility can arise as consequence of exposure to disinfectants. We also added a paper that used phenol to select for mutants with increased survival, using short exposure at high concentrations. Besides these two papers, we are not aware of other papers that evolved for improved survival upon disinfection with incomplete killing.

Text changes [in line 52-55]:

However, incomplete killing with substances used for disinfection has been shown to facilitate the selection and evolution of bacteria with improved survival to the disinfectant to which they were exposed and in addition affecting antibiotic susceptibility of the bacteria evolved to the disinfectant [1], [2].

Page 4:

Content: "antimicrobial efficacy [3]."

Comment: Again, this is another specific reference about benzalkonium chloride

Response: Most of the work regarding evolution and disinfectants has been performed with BAC. Thus, the aim of our paper was to broaden this perspective. We now cite studies dealing with other disinfectants than benzalkonium chloride. These studies investigate either the unwanted side effects of evolutionary adaptation to disinfectant or (genetic) factors that may interfere with the antimicrobial efficacy of disinfectants. [in line 57]

Page 5:

Content: "The lowest concentration which reduced the initial cell number by factor 105 was designated the MBC."

Comment: Is this the typical criterion for MBC? Can the authors provide a citation?

Response: There is no strict definition for the MBC. Most standards require disinfectants to exhibit a reduction of the initial cell number by 99.999 % to be deemed bactericidal. In line with this, the MBC here refers to a live cell reduction of 99.999 %. We have added this clarification and supporting citations to the manuscript.

Text changes: [in line 95ff]

The lowest concentration which reduced the initial cell number by 99.999 % was designated the MBC, as most standards require disinfectants to exhibit a reduction of the initial cell number by 99.999% to be deemed bactericidal [17, pp. 401–419], [18]. In line with this, the MBC here refers to a live cell reduction of 99.999%.

Page 7:

Content: "For this, the survival fraction at each transfer was normalized by the initial survival fraction. This data was then log-transformed and the area under the curve (AUC) was approximated via the trapezoidal rule."

Comment: Citation?

Response: We now provide a citation for the Trapezoidal Rule for the approximation of integrals and added the corresponding formula which was used to calculate the AUC. We now also clarify that we devised the evolution score to quantify and compare the ability to evolve towards different disinfectants.

Text changes: [in line 134 – 139]

This data was then log-transformed and the area under the curve (AUC) was approximated via the trapezoidal rule [20] according to the formula:

$$AUC = \frac{1}{2} (f(t_1) + 2f(t_2) + 2f(t_3) + \dots + f(t_n))$$

Where t_n is the n-th transfer, $f(t_n)$ is the log-transformed, normalized survival fraction at transfer t_n .

Page 7:

Content: "penalty of 1 was subtracted from the AUC for each transfer that was missed due to extinction, up to the 11th cycle."

Comment: Citation?

Response: The criterion was chosen ad hoc to account for differences in the duration of disinfection survival of populations between different compounds due to extinction of replicates. Otherwise, an early

extinct population could have a higher evolvability score as compared to a population that survives but with decreased number as compared to the start. This is now stated in the methods section.

Text changes: [in line 139-141]

An ad hoc penalty of 1 was subtracted from the AUC for each transfer that was missed due to population extinction up to the 11th cycle to account for differences in the duration of disinfection survival of populations between different compounds.

Page 8:

Content: "The goal of this study was to investigate the influence of phenotypic heterogeneity on disinfection and on the evolution of disinfectant tolerance."

Comment: Most of the information in the paragraph that begins with the sentence is not results. It's either method or it recapitulates the introduction. It should be deleted from the results.

+

Page 10:

Content: "Time-kill assays were carried out with E. coli populations in stationary phase, when persister formation is induced [6], [11], [12]."

Comment: Methods not results

+

Page 10:

Content: "To infer whether the observed time-kill kinetics were best explained by the presence of a tolerant persister subpopulation [10], a single (unimodal) and a double (bimodal) Weibull model were fitted to the data. The Weibull model has previously been used to model thermal inactivation of bacteria and can accommodate non-log-linear time-kill kinetics [14], [18], [19]. The model that best explained the observed data was selected based on goodness of fit and parsimony of the model (details in Text S1)."

Comment: This is all methods, not results

Response: We agree and we removed redundant information and moved parts of the text to the methods section.

[Line 163 to 175]

First, we investigated the phenotypic heterogeneity in survival upon disinfection by conducting time-kill assays. We used six different substances, belonging to the classes of quaternary ammonium compounds (benzalkonium chloride, BAC and didecyltrimethylammonium chloride, DDAC), alcohols (isopropanol, ISO), aldehydes (glutaraldehyde, GTA), oxidative substances (hydrogen peroxide, H₂O₂) and cationic biguanides (chlorhexidine, CHX). As a reference for the concentrations used in the time-kill assays, minimum inhibitory concentration (MIC) and minimum biocidal concentration (MBC) were determined (Table 1). The shape of the time-kill kinetics was dependent on the disinfectant (Figure 1). Fitting a single (unimodal) and a double (bimodal) Weibull model to the data showed no evidence for a persister subpopulation for H₂O₂ and GTA (Figure 1 A, B). The wide disinfection kinetics for CHX with prolonged survival for > 20 min indicated high heterogeneity of the tolerance times of individual cells (Figure 1 C). However, there was no need to invoke a persister subpopulation to explain the long-tailed time-kill curve (Table 1), as the unimodal model provided a slightly better fit (Table S1).

Page 10:

Content: "Overall, the killing with ISO proceeded faster than with the quaternary ammonium compounds BAC and DDAC."

Comment: This is true, but isn't all of this also impacted by the concentration of the antimicrobial? If lower concentrations were used, would the inactivation rate would be lower?

Response: We agree and removed the sentence.

Page 10:

Content: "MIC of colonies from the 15 minutes time-point remained unchanged compared to the reference value (Table 1)."

Comment: I don't see where the MIC of colonies from the 15 minute time point or reference values are shown in table 1.

Response: We were asked to remove a figure displaying the full result (exposed colonies vs. unexposed population) in the first round of comments. We slightly reformulated the sentence to make the result clearer. The reference value is shown in Table 1. The value of the picked colonies is not shown. However, it is unchanged; so it is the same value as shown in Table 1.

Text changes: [in line 190-193]

Resistant mutants were excluded to cause the wide tolerance time distributions with prolonged survival for > 20 min against DDAC and CHX by showing that the MIC of colonies from the 15 minutes time-point remained unchanged (data not shown) compared to the MIC displayed in Table 1.

Page 11:

Content: "To investigate the role of persistence in the evolution of disinfectant tolerance, E. coli populations were evolved under recurrent exposure to selected disinfectants with unimodal (GTA, CHX) or bimodal (ISO) time-kill kinetics."

Comment: Methods not results

+

Page 11:

Content: "Bacterial populations were exposed to disinfectant concentrations that reduced viability by 10² to 10⁴ within 2 to 15 minutes, followed by dilution and growth in fresh medium for up to 11 cycles."

Comment: Methods not results

+

Page 11:

Content: "To assess the degree of adaption in relation to the persister fraction obtained by the model (Figure 1; Table 1), evolvability scores were calculated for each population (Figure 2 F)."

Comment: As written, this sentence is methods not results

Response: We feel that these short reiterations of the performed experimental steps are required for readers to understand the context of the paragraphs stand-alone, as these are key experimental parameters important for the interpretation of the results.

Page 12:

Content: "but additional data for other disinfectants are required to further substantiate this relationship."

Comment: It would also be good to examine this with other microorganisms.

Response. We mention the need to examine our findings with other microbial species in the discussion.

Text changes: [in line: 305 - 309]

Future studies should focus on the identification of survival mechanisms to disinfectants and substantiating the here observed relation between persistence and evolvability and extend them to further disinfectant active substances and products, wider concentration ranges for a particular substance, and additional relevant microbial species.

Page 12:

Content: " *E. coli* adapted to disinfectants with bimodal time-kill kinetics but failed to adapt to disinfectants with unimodal kinetics, suggesting that in these experiments, persisters facilitated the evolution of disinfectant tolerance"

Comment: I get what the authors are saying here, but it seems to me that bimodal inactivation kinetics are evidence that a resistant subpopulation already exists. Is it possible for those situations where there's unimodal inactivation kinetics that this population does not exist because the concentration of the antimicrobial is too high. Is there any relationship between concentration of the antimicrobial and the inactivation kinetics?

Response: When choosing the concentrations, we took care to have a comparable, application-relevant degree of inactivation for all disinfectants with concentrations that inactivate at least 99.999% of the population. Because inactivation at the chosen concentration mostly occurs over the first 20 minutes (Figure 1), this approach should expose any phenotypic persister cells present. Earlier work with BAC in our lab has shown that the level of persister cells is concentration-dependent. In the literature there are several examples of concentration-dependent inactivation kinetics which we now cite. We cannot exclude that there are concentrations that would show persisters to the substances that showed unimodal inactivation kinetics at the concentrations used here and which then might (or might not) result in evolution of increased survival under these conditions. We therefore always carefully argued that the presence of persisters at the concentrations tested here facilitates tolerance evolution. We did not explicitly state that it is impossible to evolve tolerance to the compounds that did not show persisters.

We clarified this point in the discussion, added references and highlighted the need to consider the experiments with the concentrations used throughout the results and discussion sections:

Text changes:

For example: [in line 224-227]

Taken together, *E. coli* adapted to disinfectants at concentrations leading to bimodal time-kill kinetics but failed to adapt to disinfectants at concentrations leading to unimodal kinetics, suggesting that in these experiments, persisters facilitated the evolution of disinfectant tolerance.

[in line 249 257]

Importantly, we cannot exclude that there are concentrations or conditions that would also expose persistence to the substances that did show unimodal inactivation kinetics in our experiments (GTA, H₂O₂, CHX). Disinfection kinetics, and in general chemical inactivation kinetics, are concentration dependent [1], [24], [25], [26], [27]. However, the situation at or around concentrations that inactivate at least 99.999% of bacteria, as is required for disinfectants [18], is not well studied, especially not in the light of persistence and evolvability of increased survival. Therefore, future experiments should focus on these aspects to understand and prevent failure of disinfection and evolution of increased survival upon disinfection. Importantly, we cannot exclude that there are concentrations or conditions that would also expose persistence to the substances that did not show bimodal kill kinetics in our experiments (GTA, H₂O₂, CHX). Disinfection kinetics, and in general chemical inactivation kinetics, are concentration dependent [25], [26], [27], [28]. However, the situation at or around concentrations that inactivate at least 99.999% of bacteria, as is required for disinfectants [17], is not well studied, especially not in the light of persistence

and evolvability of increased survival. Therefore, future experiments should focus on these aspects to understand and prevent failure of disinfection and evolution of increased survival upon disinfection.

Page 12:

Content: "results suggest that phenotypic heterogeneity in the form of disinfectant tolerant persister subpopulations is more prevalent than previously thought. "

Comment: Previously thought by who? Can the authors supply citation?

Response: We modified the sentence.

Text changes: [in line 236-237]

Our results suggest that phenotypic heterogeneity in the form of disinfectant tolerant persister subpopulations is a prevalent phenomenon.

Page 12:

Content: "considering that persistence has been shown across a multitude of antibiotics [10], [20]."

Comment: In fact, I believe there is evidence of antibiotic resistance in populations of bacteria, recovered from samples that predate the discovery of antibiotics.

Response: We agree. However, we think it is a misunderstanding. The sentence is about phenotypic persistence, not genetic resistance.

Page 13:

Content: "Soiling can diminish the activity of disinfectants and is usually accounted for in application concentrations."

Comment: A citation here would be good.

Response: We now cite the Guidance on the Biocidal Products Regulation by the European Chemicals Agency.

Page 13:

Content: "survival of disinfection"

Comment: Survival of disinfection seems like an odd phrase. It's not the disinfection that is surviving. It is the microorganisms.

Response: We changed the wording to 'survival upon disinfection' throughout, also in the manuscript title.

Page 13:

Content: "dependent on the mode-of-action, the bacterial species, and the specificity and multitude of the cellular targets."

Comment: Probably also dependent on the concentration of the antimicrobial.

Response: We agree and modified the sentence accordingly.

Text changes: [in lines 258-260]

The mechanistic basis of prolonged survival upon disinfection is not clear but as for antibiotics it is likely multifactorial and dependent on the mode-of-action, the bacterial species, the concentration, and the specificity and multitude of the cellular targets.

Page 13:

Content: "wide disinfection kinetics"

Comment: Wide disinfection kinetics seems an imprecise phrase. Can the authors be more specific about what they mean?

Response: the term wide was specified

Text changes: For instance [in line 261-265]

Notably, all cationic substances assayed here exhibited wide disinfection kinetics with prolonged survival for > 20 min (Figure 1), which may be attributable to their specific mode of action, compared to very unspecific agents like glutaraldehyde or hydrogen peroxide for which no survival was detected after 5 min.

Page 14:

Content: " a recent study found evidence that E. faecium isolates had become more tolerant towards isopropanol after the introduction of alcohol-based hand rubs in health care facilities [34]."

Comment: While this is a correct summary of the authors findings, it is important to note that these authors used a concentration of isopropanol (23%), which is well below the recommended concentration.

Response: We added the concentration.

Text changes: [in line 293 - 296]

In line with our observation that E. coli evolved tolerance towards isopropanol, a recent study found evidence that E. faecium isolates had become more tolerant towards isopropanol at a concentration of 23 % after the introduction of alcohol-based hand rubs in health care facilities [42].

Page 15:

Content: "particular killing in terms of log reduction (typically 4 to 5 logs), but do not require full eradication [37]."

Comment: There is no such thing as "full eradication". There is only inactivation to below detection limits. This depends on the starting concentration, the inactivation kinetics, and the detection limit.

Response: We agree that full eradication cannot be determined in practice. We modified the sentence accordingly.

Text changes: [in line 317-320]

Such incomplete eradication is covered by international standards that define the appropriate efficacy of disinfectants to ensure a particular killing in terms of log reduction (typically 4 to 5 logs) [17], but do not require eradication below the lowest possible limit of detection (i.e. close to full eradication) [45].

Page 15:

Content: "Safeguarding the efficacy of disinfectants requires"

Comment: One commonly used method to prevent development of resistance is to change disinfectants to another disinfectant with a different mode of action. Therefore, it would also be useful to study how sequential exposure to different disinfectant impacts the development of resistance evolution.

Response: We thank the reviewer for this thought. We added the thought to the sentence.

Text changes: [in line 326-330]

Safeguarding the efficacy of disinfectants requires further research into more bacterial species, the physiology of disinfectant persisters, their occurrence under more realistic disinfection scenarios, their role for evolvability of increased survival, the physiology of evolved disinfectant tolerant mutants also in terms of antibiotic susceptibility, how cycling of disinfectants impacts evolution, and potential barriers for evolution in the field.

Re: Spectrum03276-22R1 (Heterogeneous survival of disinfection underlies evolution of increased tolerance)

Dear Dr. Frank Schreiber:

Your manuscript has been accepted, and I am forwarding it to the ASM production staff for publication. Your paper will first be checked to make sure all elements meet the technical requirements. ASM staff will contact you if anything needs to be revised before copyediting and production can begin. Otherwise, you will be notified when your proofs are ready to be viewed.

Sincerely,
Olaya Rendueles
Editor
Microbiology Spectrum